# Surface Immunogenic Protein of *Streptococcus* Group B is an Agonist of Toll-Like Receptors 2 and 4 and a Potential Immune Adjuvant

**DOI:** 10.3390/vaccines8010029

**Published:** 2020-01-16

**Authors:** Diego A. Diaz-Dinamarca, Ricardo A. Manzo, Daniel A. Soto, María José Avendaño-Valenzuela, Diego N. Bastias, Paulina I. Soto, Daniel F. Escobar, Valeria Vasquez-Saez, Flavio Carrión, Magdalena S. Pizarro-Ortega, Christian A. M. Wilson, Julio Berrios, Alexis M. Kalergis, Abel E. Vasquez

**Affiliations:** 1Seccion de Biotecnologia, Instituto de Salud Publica de Chile, Santiago 7780050, Chile; diego.d.dinamarca@gmail.com (D.A.D.-D.); rmanzo@ispch.cl (R.A.M.); dasoto@ispch.cl (D.A.S.); mariajoavendanov@gmail.com (M.J.A.-V.); dgo.basta@gmail.com (D.N.B.); paulinasoto.bqa@gmail.com (P.I.S.); descobar@ispch.cl (D.F.E.); valeria.vasquez-saez@gmail.com (V.V.-S.); 2Millenium Institute on Immunology and Immunotherapy, Departamento de Genética Molecular y Microbiología, Facultad de Ciencias Biológicas, Pontificia Universidad Católica de Chile, Santiago 8320000, Chile; mfpizarro@uc.cl (M.S.P.-O.); akalergis@bio.puc.cl (A.M.K.); 3Escuela de Biotecnología, Facultad de Ciencias, Universidad Santo Tomas, Santiago 8320000, Chile; 4Programa de Inmunología Traslacional, Facultad de Medicina, Clínica Alemana Universidad del Desarrollo, Santiago 8320000, Chile; fcarrion@udd.cl; 5Departamento de Bioquímica y Biología Molecular, Facultad de Ciencias Químicas y Farmacéuticas, Universidad de Chile, Santiago 8320000, Chile; yitowilson@gmail.com; 6Escuela de Ingeniería en Bioquímica, Pontificia Universidad Católica de Valparaíso, Valparaíso 2340000, Chile; julio.berrios@pucv.cl; 7Departamento de Endocrinología, Facultad de Medicina, Pontificia Universidad Católica de Chile, Santiago 8320000, Chile; 8Facultad de Ciencia, Universidad San Sebastián, Providencia, Santiago 8320000, Chile

**Keywords:** surface immunogenic protein, group B *Streptococcus*, TRL2 and TLR4 agonist, adjuvant protein

## Abstract

Vaccine-induced protection against pathogens, especially subunit-based vaccines, are related to antigen properties but mainly in their ability to stimulate the immune system by the use of an adjuvant. Modern vaccines are formulated with a high level of antigen purity, where an efficient adjuvant is necessary. In this context, the use of protein Toll-Like Receptor (TLR) agonists as vaccine adjuvants has been highlighted because of their optimal immunogenicity and minimal toxicity. The Surface Immunogenic Protein (SIP) from Group B *Streptococcus* (GBS) has gained importance as a new potential protein-based vaccine. Recently, we reported that recombinant SIP (rSIP) expressed by E. coli and purified by High Performance Liquid Chromatography (HPLC) alone induces a protective humoral immune response. In this study, we present the immunomodulatory properties of rSIP as a protein-based adjuvant, as an agonist of TLR. To this end, we showed that C57BL/6 bone marrow-derived dendritic cells pulsed by rSIP resulted in enhanced CD40, CD80, CD86, and Major Histocompatibility Complex (MHC) class II as well as increased secretion proinflammatory cytokines Interleukin (IL)-6, Interferon (IFN)-γ, Tumor Necrosis Factor (TNF)-α, and IL-10. Next, we investigated the in vivo effect of rSIP in the absence or presence of ovalbumin (OVA) on antigen-specific antibody secretion in C57BL/6 mice. Immunization with rSIP plus OVA showed that anti-OVA IgG2a and IgG1a increased significantly compared with OVA alone in C57BL/6 mice. Also, the immunization of rSIP plus OVA generates increased serum cytokines levels characterized by IL-12p70, IL-10, IL-4, and IFN-γ. Interestingly, we observed that rSIP stimulate Toll Like Receptor (TLR)2 and TLR4, individually expressed by Human embryonic kidney (HEK) 293-derived TLR reporter cells. These findings suggest that rSIP is a new potential protein TLR agonist adjuvant and may be employed in the development of new vaccines.

## 1. Introduction

The majority of new human vaccines are based on purified antigens, which generally have low immunogenicity, and adjuvants are necessary to improve vaccine-induced immune responses [1,2]. Due to their role in self/nonself-differentiation and their ability to induce professional antigen-presenting cell (APC) maturation and to subsequently trigger stronger immune responses, Toll-Like Receptor (TLR) agonists are considered promising adjuvant candidates [2,3]. In this regard, the majority of the currently investigated TLR agonists are nonprotein microbial components such as lipopolysaccharides, oligonucleotides, and lipopeptides [4]. However, protein TLR agonists are moving forward because of their high immunogenicity and minimal toxicity. Moreover, protein adjuvants can be genetically fused to protein antigens, ensuring the co-delivery of adjuvant antigens, leading to more effective activation of the innate and adaptive immune responses [3,4].

TLR2 and TLR4 have gained importance due to their extreme ability to identify distinct molecular patterns from invading pathogens and to exhibit several core properties of vaccine adjuvants [4]. The engagements of TLR2 and TLR4 induce an innate immune response and proinflammatory cytokines in vitro and in vivo and induces co-stimulatory markers on macrophages and dendritic cells (DCs) [2,4]. Studies have highlighted properties of microbial protein TLR agonists showcasing immunomodulatory properties that parallel the adjuvant activity. For example, the outer membrane proteins (OMPs) of *Shigella flexneri* [5] and Lumazine synthase from *Brucella spp.* (BLS) [6] are known to induce TLR2 and TLR4 signaling, respectively. In this situation, a new protein TLR agonist could be considered an attractive immunotherapeutic vaccine against cancer and can potentially enhance immune responses of vaccines in the elderly, pregnant women, and immuno-compromised populations [4].

The Surface Immunogenic Protein (SIP) from Group B *Streptococcus* (GBS) is an immunogenic and conservative antigen in all GBS serotypes. The subcutaneous, intranasal, and oral immunization with recombinant SIP (rSIP) elicited specific opsonophagocytic antibodies that confer protection against GBS [7]. Also, this protein stimulates a cellular and humoral immune response [8,9]. In this study, we showed that C57BL/6 bone marrow-derived DC (BM-DC) stimulated with rSIP resulted in enhanced co-stimulatory proteins as well as enhanced production of proinflammatory cytokines IL-6, IFN-γ, TNF-α, and IL-10. Furthermore, immunization with rSIP plus ovalbumin (OVA) showed that anti-OVA IgG2a and IgG1a increased significantly in comparison with OVA alone in C57BL/6 mice. In addition, immunization of rSIP plus OVA generates increased serum cytokines levels characterized by IL-12p70, IL-10, IL-4, and IFN-γ. Finally, rSIP can activate TLR2 and TLR4 HEK293 blue reporter cells.

## 2. Materials and Methods

### 2.1. Ethics Statement

All the experiments that used mice were conducted in agreement with the international ethical standards and followed the Chilean Law 20380 on Animal Protection (2009). The experimental protocol was reviewed and approved by the Institutional Committee of Care and Use of Laboratory Animals (CICUAL) of Instituto de Salud Publica de Chile. 

### 2.2. Mice Strains

Seven-to eight-week-old C57BL/6 wild-type (WT) female mice were originally purchased from Jackson Laboratories (Bar Harbor, ME, USA) and maintained in the pathogen-free animal facility at the Instituto de Salud Pública de Chile.

### 2.3. Purification of Recombinant SIP

The gene from the surface immunogenic protein (GenBank accession no: KX363665.1) was cloned from the GBS bacterial strain serotype III (GenBank accession no: KU736792.1), following the protocol described in Diaz-Dinamarca et al., [7]. Briefly, recombinant SIP (rSIP) was expressed in *Escherichia coli* BL21 (DE3) codon and previously transformed with the plasmid pET21a::sip. The rSIP was expressed as a soluble protein and purified using nickel-nitrilotriacetic acid (NI-NTA) resin by low-pressure chromatography and High Precision Liquid Chromatography (HPLC) using a molecular exclusion column. Purified rSIPs were Lipopolysaccharides (LPS)-free after HPLC purification step.

As a control for TLR cell-based assay, the same SIP gene was cloned into the pPICZα vector and expressed in Pichia pastoris using methanol as inducer. rSIP was secreted into *P. pastoris* culture as a soluble protein and purified by Ni-NTA resin by low-pressure chromatography [10]. Finally, rSIP expressed in *E. coli* and *P. pastoris* was analyzed by SDS-PAGE and Western blot using a polyclonal antibody against the SIP available in our laboratory. The recombinant proteins were quantified by the Bicinchoninic Acid (BCA) method.

### 2.4. Circular Dichroism Measurements

In order to analyze the secondary structure of rSIP purified from *E. coli*, circular dichroism spectroscopy was performed using a No. J-810 instrument (JASCO, Essex, UK). The spectra were measured in the far ultraviolet region, from 260 to 190 nm. Optical path length cuvettes at 0.1 mm were employed. Each spectrum was obtained from the accumulation of at least three scans at working temperature.

### 2.5. Animal Immunization

To evaluate the ability of rSIP of GBS purified from *E. coli* to induce an antigen-specific immune response in a mouse model against ovalbumin (OVA) protein, we used rSIP as an adjuvant for immunization with OVA. For stimulation of secretion of anti-OVA antibodies, C57BL/6 mice (five per group) were subcutaneously immunized four times with 100 µL of PBS-1X, OVA (10 µg), OVA (10 μg)/ Imject™ Alum Adjuvant (alum; 2 mg), and OVA (10 μg)/rSIP (10 μg). Animal immunization schedule was performed as previously described by Liu et al., [11]. The female mice were immunized four times: on days 1, 14, 28, and 42 of the experiment. The experimental animals were euthanized five days post last immunization with a lethal dose of i.p. anesthetic (120 mg/kg ketamine; 10 mg/kg xylazine).

### 2.6. Measurement of Anti-OVA Specific Antibodies

ELISA was performed according to a modified procedure, as previously described by Yanese et al., [12]. ELISA determined OVA-specific total IgG, IgG1, and IgG2a in the mouse sera. The sera from preimmunized mice were used as a negative control. After coating plates with 1 μg of OVA, serial dilutions of mice sera were added and incubated. The specific level of immunoglobulins was detected using alkaline phosphatase-conjugated secondary goat anti-mouse-IgG, -IgG1, and -IgG2a (R&D Systems). The immunoglobulin level corresponded to 1:25 dilution and is expressed by absorbance units at 450 nm. 

### 2.7. Bone Marrow Dendritic Cell Culture

Bone marrow-derived DCs (BM-DCs) were cultured using the method described by Lutz et al., [13], from female C57bl/6 mice. Briefly, on day 0, femurs and tibia of the mice were flushed and the resulting bone marrow suspension was passed through a 70-μm cell strainer (BD Biosciences) to obtain a single-cell suspension. Red blood cells were subsequently lysed using Pharm Lyse Lysing Buffer (BD Biosciences). Cells were seeded at 1 × 10^6^ cells per mL of Roswell Park Memorial Institute (RPMI) medium with L-glutamine (Invitrogen) supplemented with 10% heat-inactivated fetal calf serum (Gibco), 2% hydroxyethyl piperazineethanesulfonic acid (HEPES), 1% penicillin-streptomycin, and 40 ng/mL of murine granulocyte-macrophage colony-stimulating factor (rGM-CSF; BD Pharmingen). Then, cells were incubated at 37 °C and 5% CO_2_. On day 3, a fresh RPMI medium containing rGM-CSF was added to double the original volume. On day 5, the dendritic cells were pulsed for 24 h with rSIP (10 μg/mL). LPS (50 ng/mL) and PBS-1X were used as positive and negative controls, respectively.

### 2.8. Flow Cytometry Analysis of BM-DC Phenotypic Markers

Following the 24 h of stimulation of BM-DCs and rSIP, the expressions of CD11c, Major Histocompatibility Complex (MHC) class II, CD80, CD86, and CD40 were analyzed by flow cytometry. Cells were incubated for 30 min at 4 °C in the dark, with antibodies against CD11c-BV421, I-A⁄I-E-APC, CD80-PerCP-Cy5.5, CD40-Fluorescein IsoTioCyanate (FITC), and CD86-PE (BD-Pharmingen) in the presence of Fixable Viability Stain 510 (BD Horizon) to discard dead cells. Flow cytometry data were analyzed, and mean fluorescent intensities of CD80, CD86, CD40, and MHC-II expressions of viable (exclusion of dead cells) CD11c+ gated cells were also analyzed. The acquisition was performed with the FACSVerse flow cytometer (BD Biosciences) and analyzed with Flow Jo software (Tree Star, USA).

### 2.9. Measurement of Serum Cytokine Profiles

Levels of cytokines IL-4, IL-10, IFN-γ, and IL-12p70 in the mouse serum were quantitatively determined by ELISA using Opt-EIA kit (BD Biosciences) specific for each cytokine according to manufacturer’s instructions. Briefly, sera from the immunized mice were added to the cytokine-specific antibody-coated 96-well plates and incubated at 37 °C for 1.5 h. After removal of unbound serum proteins, a biotin-labeled cytokine-specific detection antibody was added and incubated at 37 °C for 1.5 h. The plates were washed and incubated with streptavidin- Horseradish Peroxidase (HRP) for 30 min. After washing, the 3,3′,5,5′-Tetramethylbenzidine (TMB) substrate solution was added and the color reaction was developed. The absorbances were read at 450 nm using a microplate reader and then used to calculate the concentration (pg/mL) against the standard curve.

### 2.10. Detection of Soluble Cytokines Supernatants of BM-DCs

The supernatant from BM-DCs pulsed with rSIP from *E. coli*, LPS, and PBS-1X was collected at 24 h to analyze soluble cytokines. The quantitative determination of inflammatory cytokines in serum was performed using the Cytometric Bead Array (CBA) Mouse Th1/Th2/Th17 Cytokine kit (BD Biosciences). This kit allows to quantitatively measure interleukin (IL) 2, IL-4, IL-6, and IL-10; Tumor Necrosis Factor (TNF) α; Interferon (IFN) γ; and IL-17A protein levels in a single sample. The fluorescence produced by CBA beads was measured on a FACSVerse flow cytometer (BD Biosciences) and analyzed using FCAP array software (Soft Flow Inc). The limits of detection of each cytokine are 0.1 pg/mL (IL-2), 0.03 pg/mL (IL-4), 1.4 pg/mL (IL-6), 0.5 pg/mL (IFN-γ), 0.9 pg/mL (TNF-α), 0.8 pg/mL (IL-17A), and 16.8 pg/mL (IL-10).

### 2.11. Secreted Alkaline Phosphatase Nuclear Factor Kappa-Light-Chain-Enhancer of Activated B Cells (NF-κB) Activity Assays

HEK blue mTLR2 and HEK blue mTLR4 cells were obtained from InvivoGen. The cells were cultured in Dulbecco’s Modified Eagle Medium (DMEM), 4.5 g/L glucose, 10% (*v*/*v*) fetal bovine serum, 50 U/mL penicillin, 50 mg/mL streptomycin, 100 mg/mL Normocin™, and 2 mM L-glutamine at 37  °C in humidified air containing 5% CO_2_ as per manufacturer’s instructions. The HEK blue mTLR2 and HEK blue mTLR4 cells were seeded into 96-well plates at a density of 5 × 10^4^ cells/well. rSIP purified by HPLC, obtained from *E. coli* in HEK blue detection solution, was added to the cells to produce a final concentration of 50 ng/µL. After 24 h incubation at 37 °C, the optical density (OD) of the samples was measured at 620 nm using a microplate reader. For each set of experiments, control of free-endotoxin protein using rSIP from *E. coli* boiled for 15 min and a negative test control using rSIP expressed by *Pichia Pastoris* were also performed. As a positive control, the specificity of HEK293-derived TLR reporter cells was performed using an additional stimulation with lipopolysaccharide from *E. coli* K12 (100 ng/mL, InvivoGen, USA) and peptidoglycan Pam3CSK4 (300 ng/mL, Sigma Aldrich, USA).

### 2.12. Cytotoxicity and Cell Viability Assay

Annexin V: Cells were seeded into 12-well plates at a density of 2 × 10^6^ cells/well and treated with the following stimuli for 24 h: PBS-1X, LPS (1 and 10 μg/mL), and rSIP (0.1, 1, 10, and 30 μg/mL). The cells were washed with PBS-1X and stained with the Annexin V/PI apoptosis detection kit (BD Bioscience) following manufacturer’s instructions.

Active caspase 3: Cells were washed, fixed, permeabilized, and stained with FITC-conjugated mouse monoclonal anti-active-caspase-3 mAb according to manufacturer’s instructions (BD, #550480) and analyzed on FACSVerse flow cytometer (BD Biosciences).

### 2.13. Statistical Analysis

The results are presented as the mean and standard deviation. Shapiro–Wilk test was used to evaluate the normality of the distribution of the examined variables. The statistical data analysis was performed using the Student’s *t*-test and ANOVA test. *p*-values < 0.05 were considered statistically significant. The analyses were performed using GraphPad Prism software (GraphPad Software, Inc., USA).

## 3. Results

### 3.1. The Surface Immunogenic Protein of GBS Forms a Homodimer with a Principal β-Sheet Secondary Structure

To gain insight into the structure of rSIP from GBS, we purified the recombinant protein by low- and high-pressure liquid chromatography and analyzed the secondary structure of our protein. SIP was analyzed by Size Exclusion Chromatography (SEC) by HPLC, and a heterodimer of 93 kDa was observed. We dissociated the protein in the presence of 6M urea as a monomer control (Figure 1A). The high purity of the protein allows us to describe that, principally, the protein contains a β-sheet secondary structure by circular dichroism (Figure 1B).

### 3.2. The Surface Immunogenic Protein of GBS Increases Immunoglobulin Secretion Against OVA Protein

Upon closer protein characterization of rSIP, as subcutaneous and oral immunization of the protein generates a protective immune response against GBS [7,8,9], it is hypothesized that it could be a vaccine adjuvant. To evaluate the rSIP ability to stimulate an immune response against another antigen, we evaluated immunoglobulin levels in mice immunized with rSIP + OVA. The immunization scheme is described in Figure 2A. Subcutaneous immunization with rSIP + OVA increases the levels of IgG, IgG1, and IgG2a anti-OVA in comparison with immunization with OVA protein alone, indicative of Th1/Th2-biased systemic humoral immunity. As a positive control, a control group immunized with OVA + Alum generated antibody levels similar to rSIP + OVA (Figure 2B–D). Serum cytokine profiles also showed a Th1/Th2 bias in mice immunized with rSIP + OVA, with significantly higher levels of IFN-γ, IL-12p70, IL-4, and IL-10 in comparison with the PBS-1X control group (Figure 2E).

### 3.3. rSIP of GBS Induces Maturation of Murine Bone Marrow-Derived DCs

The maturation of BM-DCs plays a crucial role in mediating immune responses to infection. In the presence of GM-CSF for 6 days, cells from the bone marrow of C57BL/6 female mice form a culture to induce an immature phenotype of BM-DCs. Upon encounter with microbial, proinflammatory, or T cell-derived stimuli, maturation of DC is induced, characterized by phenotypic and functional changes. Mature BM-DCs exhibit reduced phagocytic, increase expression of MHC and co-stimulatory molecules, and secrete large amounts of immunostimulatory cytokines. The mature BM-DCs have morphological changes, observing a cluster aggregation growth (data not shown). Therefore, to investigate whether rSIP indicated BM-DCs maturation, we measured the expressions of CD80, CD40, CD86, and MHC-II by flow cytometry as described in the Materials and Methods section. Expressions of CD80, CD40, CD86, and MHC-II molecules increased in a dose-dependent manner on BM-DCs stimulated with rSIP (5 µg/mL and 10 µg/mL) (Figure 3). As a positive control, BM-DCs were stimulated with LPS (5 µg/mL). As a negative control, untreated BM-DCs retained an immature phenotype.

### 3.4. rSIP Promotes the Secretion of Proinflammatory Cytokines from BM-DCs

DC-derived cytokines play a critical role in the polarization of T-cells and in mediating inflammatory responses [14]. To determine whether rSIP affects the production of pro-inflammatory cytokines from DCs, we treated BM-DCs from C57BL/6 mice with rSIP expressed on *E. coli* and purified by HPLC (Figure 4A). The levels of IL-6, IFN-γ, IL-10, and TNF-α in rSIP-treated BMDCs were significantly higher in comparison with the PBS-treated BM-DCs (Figure 4B–E). The levels of IL 2, IL-4, and IL-17A showed no significant difference to the PBS-1X control group (data not shown). The results showed that rSIP promotes the production of proinflammatory cytokines from BM-DCs at both concentrations of 1 µg/mL and 10 µg/mL. As a positive control, we used LPS (10 µg/mL). These findings suggested that rSIP was capable of inducing functional maturation of BM-DCs.

### 3.5. Activation Induced Cell Death upon rSIP Stimulation

As maturation of BM-DCs could be associated to cell death, we investigated whether rSIP induces cytotoxicity of BM-DCs by analyzing cell viability, cell death, and caspase-3 activity. Based on these results, we found that the proper concentration of rSIP was 0.1 μg/mL, which did not register toxicity levels. Then, analyzing the proinflammatory activity of rSIP-treated BMDCs, we confirmed the cytotoxic effect of our protein at a concentration above 1 μg/mL in comparison with BM-DCs treated with PBS-1X and showed that BM-DCs had no cytotoxic effect in the presence of 0.1 μg/mL of rSIP for 24 h (Figure 4F). BM-DCs co-treated with rSIP led to increased percentages of necrosis and apoptosis cells and a decreased cell viability (Figure 4G). To identify whether apoptosed BM-DC cells were associated to caspase-3, BMDCs were pulsed with increased concentrations of rSIP for 12 h. Apoptosis of BM-DCs pulsed with rSIP was not associated to caspase-3 activation (Figure 4H). These results indicate that rSIP is nontoxic at concentrations below 0.1 μg/mL and that rSIP can be associated to BM-DC necroptosis, a form of cell death known to involve inflammasome activation and IL-1β secretion, which proceeds in a caspase-independent manner [15].

### 3.6. rSIP Stimulates HEK Blue TLR2 and TLR4

TLR2 and TLR4 have gained much attention due to their extreme ability to identify diversified ligands [16]. In order to determine if rSIP could activate an innate immune response, we used HEK blue-2 and -4 cells to study the activation of TLR2 and TLR4 by monitoring NF-kB activation. Activation of NF-κB was measured by the detection of secreted alkaline phosphatase (SEAP) that is under the control of the NF-κB promoter. Therefore, NF-κB activation by TLR2 and TLR4 leads to SEAP secretion, which is detected by an alkaline phosphatase substrate in cell culture media [17]. rSIP was used on 50 ng/mL to activate HEK blue-2 and -4 cells. LPS (100 ng/mL) and PAM3CK4 (300 ng/mL) were used as a positive control. Normal media were used as a negative control to ensure possible endogenous alkaline phosphatases. rSIP stimulates HEK blue-TLR2 and -TLR4 (Figure 5A,B). Also, as protein control, we used a denaturant rSIP (95 °C for 15 min) and rSIP obtained from *P. pastoris*.

## 4. Discussion

GBS is a leading cause of young infant mortality and morbidity globally, with vaccines being developed for over four decades but none licensed to date [18]. The subcutaneous immunization with rSIP alone generates a decrease in GBS vaginal colonization in a murine model and protective antibodies [8,9]. Due to the immunogenic potential of rSIP, we evaluated the capacity of rSIP to generate an immune response as a vaccine adjuvant. In this study, we purified and characterized the adjuvant capacity of rSIP from GBS expressed by *E. coli*. The high purity level of rSIP was used to evaluate the effect on OVA-specific immunoglobulin, the maturation and proinflammatory cytokine profile of BM-DCs, and the stimulation of HEK blue TLR2 and TLR4 cells.

Different ligands from organisms such as *Pseudomonas aeruginosa, Plasmodium falciparum, Toxoplasma gondii, Leishmania major*, and *Entamoeba histolytica* have been described as ligands for TLR2 and TLR4 [19]. To date, no model or crystal structure shows the mechanism of TLR2 and TLR4 signaling by proteinaceous ligands. Structure–activity relationships could determine TLR binding and consequent stimulation of the innate immune response, which has been investigated for a range of lipopeptides [20]. Details of the characterization of HEK blue-4 (TLR4 reporter) and HEK blue-2 (TLR2 reporter) cells as well as their ligand specificities to various TLR4 and TLR2 have been described and previously validated by Hood et al., [21]. The tertiary structure and the level of posttranscriptional modifications of rSIP conditioned the stimulation of HEK blue-2 (TLR2 reporter) and HEK blue-4 (TLR4 reporter) cells (Figure 5). These notions are correlated with the denatured protein, which cannot activate any TLR2 and TLR4 HEK blue cells, and rSIP from *P. pastoris* only activates TLR4 reporter cell. This approach indicates that rSIP has a conformational epitope that could be essential for vaccine adjuvant activity. In this context, it has been estimated that most B-cell epitopes (up to 90%) are conformational [22] and could explain an increased level of anti-OVA antibodies in mice immunized with rSIP as vaccine adjuvant: an interesting outcome in the development of rSIP as a possible vaccine adjuvant and/or vaccine against GBS. Despite the success of TLR2 and TLR4 stimulated by rSIP and its potential as a vaccine adjuvant, one of the plausible limitations is to evaluate the expression of co-stimulatory molecules necessary for the activation of naïve T-cells using rSIP TLR agonists. On the other hand, future improvements in obtaining rSIP could involve the use of *P. pastoris*, the co-delivery of rSIP antigens, and encapsulation for improved delivery as a vaccine. Vaccine safety and the transition from whole-pathogen vaccines to protein-subunit vaccine technologies require the development of new vaccine adjuvants to boost immunogenicity [23]. 

In vaccination, the adjuvant’s nature contributes to modifying proinflammatory properties. High levels of antibodies and isotype class switching toward IgG1 are considered features of the Th2-type immune response, along with anti-inflammatory cytokines IL-4 and IL-10 [24]. Such responses are classically induced by alum [25] and TLR2 adjuvants [26]. In contrast, the production of IgG2a, IgG2b, IgG2c, and IgG3 as well as proinflammatory cytokines IFN-γ, IL-2, and IL-12 indicates induction of Th1-type immune responses, favored by TLR4 and TLR9 adjuvants [27]. In this study, alum plus OVA and rSIP plus OVA induce OVA-specific IgG1 and IgG2a antibodies, which are important for complement-independent pathogen neutralization and complement fixation, respectively. Also, IgG1 and IgG2a are complement- and Fc-mediated bacterial opsonophagocytosis. Although the signaling pathways of TLR2 and TLR4 converge to the same adaptor protein myeloid differentiation factor 88 (MyD88) [28], their cytokine production patterns after stimulation are different. Whereas the mechanism of TLR4-induced IFN-γ secretion via IL-12 has been described [29], the pathway of TLR2-induced involved more Th2 cytokine patterns via IL-12 suppression [28]. Our study indicated that rSIP is a TLR2 and TLR4 agonist that could explain the systemic production of IL-4, IL-10, IL-12p70, and IFN-γ in the serum from mice immunized with rSIP + OVA. Also, based on the fact that rSIP is an agonist of TLR2 and TLR4, it states that it is capable of promoting a balanced Th1/Th2 immune response. Our studies thus provide the first direct evidence of TLR2- and TLR4-dependent activity for rSIP.

In general, DC maturation enhances their antigen presentation capacity and ability to activate T-cells and is a prerequisite for the induction of potent and long-lasting immunity, crucial at the interface between innate and adaptive immune responses [2]. Adjuvants generally act by activating innate and adaptive immune responses through BM-DC maturation. In this study, rSIP induced maturation of bone marrow-DCs, supported by changing morphology (data not shown) and expression of cell surface markers. Also, rSIP induces the secretion of proinflammatory cytokines consisting of IL-6, IFN-γ, IL10, and TNF from the mature BM-DCs, which is essential to regulate the TLR-induced Th1/Th2 immune response [30]. Programmed cell death in DCs is essential for regulating DC homeostasis and, consequently, the scope of immune responses. Maintenance of DC homeostasis by programmed cell death has major impacts on the scope of antigen-specific immune responses and immune tolerance. It has been seen that exposure to proinflammatory cytokines could induce DC cell death [31]. In line with this observation, rSIP from HPLC purified *E. coli* generates cell death at high concentrations of protein characterized by a possible unidentified mechanism of necroptosis. The relationship of cell maturation by rSIP and programmed cell death should be studied in more detail due to the over-accumulation or depletion of DCs that may disrupt immune tolerance.

The role of TLR in vivo bacterial infections has been studied in TLR-deficient mice. *Salmonella typhimurium*, a Gram-negative bacterium that can replicate in macrophages, has at least four Pathogen-Associated Molecular Patterns (PAMPs) detected by TLRs: lipoprotein (TLR2), LPS (TLR4), flagellin (TLR5), and CpG-DNA (TLR9) [4,19]. In the case of TLR2, interaction with lipoproteins from GBS is an important outcome for sepsis development [32]. Conversely, it has been described that TLR4 is not implied in the proinflammatory responses by GBS in macrophage from mice [32]. TLR2 and TLR4 are expressed in various immune cells, including neutrophils, monocytes/macrophages, and DCs. Amongst these, neutrophils first migrate to the site of infection, sense the pathogen, and elicit an immune response. However, coordinated activation of adaptive response is mediated through the binding of a specific ligand to monocytes or DCs that are also mediated principally by TLR2 and TLR4. Moreover, these TLRs are also expressed on classical adaptive immune cells like B and T lymphocytes [19]. This property correlates with the previous report that rSIP as a vaccine against GBS generates neutrophil recruitment on the site of infection, a cellular immune response against the bacterial infection, and functional opsonic antibodies [8,9]. 

## 5. Conclusions

Our results are a novelty in describing the first GBS ligand for TLR2 and TLR4 and could be beneficial in the development of a vaccine adjuvant because TLR4 ligand-based adjuvants are the most advanced in commercial vaccines [33] and because TLR2 adjuvant has attracted great interest as an efficient adjuvant for vaccines against infectious diseases [34]. In this study, we prove that rSIP has immunostimulatory properties, which could be used as a vaccine adjuvant. This observation correlates with the notion that subunit vaccines have been considered to be safe and are recommended to protect the pregnant woman, fetus, and infant from vaccine-preventable diseases [35]. In this context, a plausive new vaccine associated with pregnancy could take advantage of the adjuvant, and the protective capacity of rSIP, for example, the co-delivery of the nucleoprotein from respiratory syncytial virus [36] plus rSIP, could protect infants against both diseases. Finally, considering that rSIP immunogenicity studies are in the advanced preclinical stage, this protein is a candidate for a potential new vaccine adjuvant.

## Figures and Tables

**Figure 1 vaccines-08-00029-f001:**
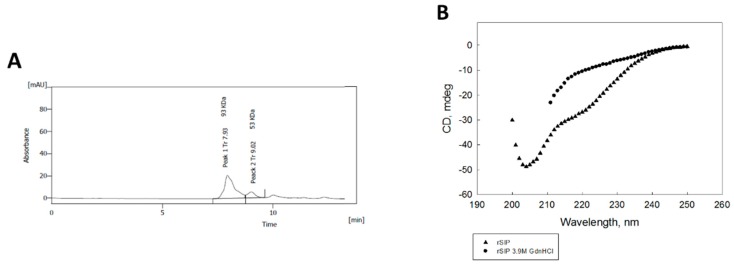
Circular dichroism spectra of Recombinant Surface Immunogenic Protein (rSIP) with a representative β-sheet secondary structure: (**A**) HPLC analysis of purified rSIP under 6 M of urea. Size Exclusion Chromatography (SEC) was performed on purified rSIP under 6M urea. Peak 1 corresponds to an aggregated form of rSIP (93 kDa, dimer), and peak 2 corresponds to a monomer form of rSIP. PSS-pro kit (PSS GmbH, Germany) was used as a standard curve for molecular mass. (**B**) Circular dichroism spectroscopy was performed using a model No J-810 instrument. The spectra were measured in the far ultraviolet region from 260 to 190 nm. As denaturation control, the rSIP (10 µM) was incubated in 3.9M GdnHCl for 2 h at 4 °C.

**Figure 2 vaccines-08-00029-f002:**
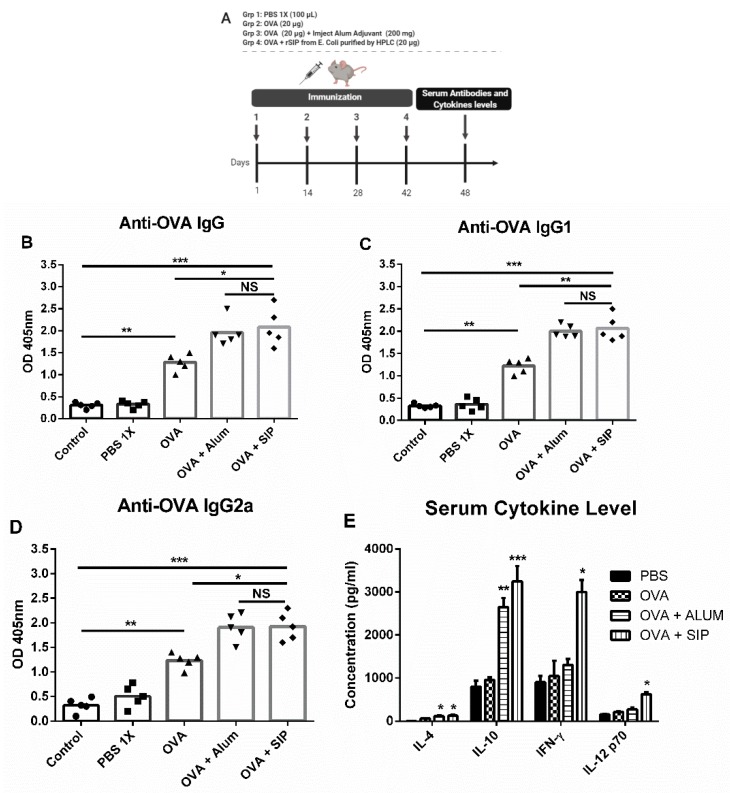
Effect of rSIP on the induction of antibody levels and cytokine in sera: (**A**) Schematic representation of the immunization used in the C57BL/6 mice experiment. The mice (*n* = 5) were subcutaneously immunized four times with 100 µL of PBS-1X, ovalbumin (OVA) (10 µg), OVA (10 μg)/alum (2 mg), and OVA (10 μg)/rSIP (10 μg). Serum samples were collected and diluted for detection by ELISA on day 6 post-final boost. OVA at 2 μg/mL was coated on each well in 96-well plate. Serum anti-OVA: (**B**) IgG, (**C**) IgG1, and (**D**) IgG2a antibody absorbance levels were determined over the pre-immune serum (control). Results represent 1 of 2 independent experiments with similar results. *** *p* < 0.0001; ** *p* < 0.001; * *p* < 0.05 by ANOVA multiple comparisons for OVA/rSIP-immunized mice compared with OVA-, PBS-1X-, and pre-immune serum of mice (control). Cytokine levels in sera of immunized animals: (**E**) The levels of Th1 cytokines (IFN-γ and IL12) and Th2 cytokines (IL-4 and IL-10) in sera from immunized mice were detected by quantitative ELISA. Results represent 1 of 2 independent experiments with similar results. Data are shown as mean ± SD of five mice in each group. The asterisk indicates a significant difference versus PBS control (not significant (ns); ** *p* < 0.01; *** *p* < 0.001; **** *p* < 0.0001).

**Figure 3 vaccines-08-00029-f003:**
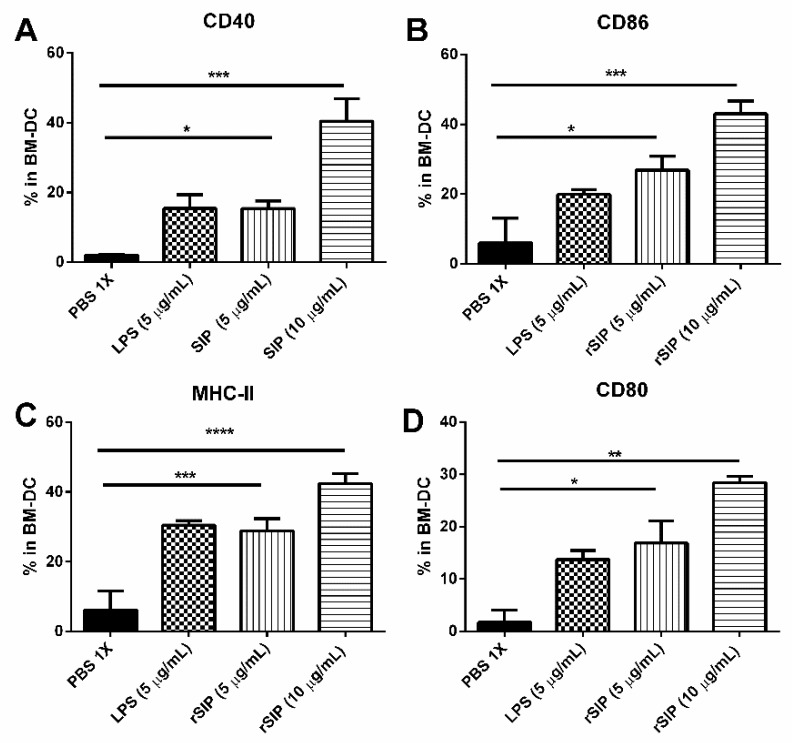
rSIP promotes bone marrow-derived dendritic cell (BM-DC) maturation. Hematopoietic progenitor cells from the bone marrow of C57BL/6 mice were cultured in the presence of granulocyte-macrophage colony-stimulating factor (GM-CSF) for six days. Then, on day 6, BM-DCs were stimulated with 5 μg/mL and 10 μg/mL of rSIP for 24 h or with LPS (5 μg/mL) as a positive control. Also, PBS-1X served as a negative control. (**A**) CD40, (**B**) CD86, (**C**) Major Histocompatibility Complex (MHC)-II, and (**D**) CD80 surface markers were analyzed by flow cytometry. The values shown in the flow cytometry profiles are the mean fluorescence intensity (MFI) indexes (**** *p* < 0.0001; *** *p* < 0.001; ** *p* < 0.01; * *p* < 0.05 by ANOVA multiple comparisons for rSIP stimulation compared with PBS-1X).

**Figure 4 vaccines-08-00029-f004:**
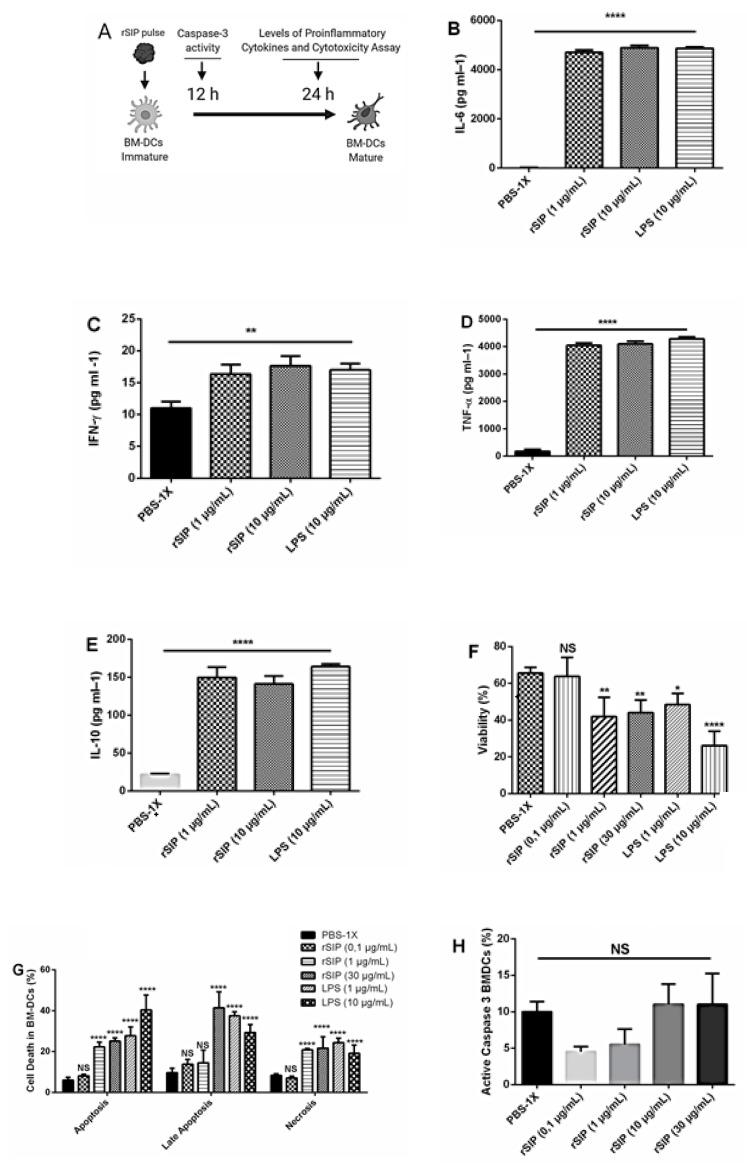
rSIP induces proinflammatory cytokine profile and cell death independent of active caspase-3 in BM-DCs. After BM-DC differentiation for six days, BM-DCs were stained with anti-CD11c and Annexin V and PI Abs, and then, cells were analyzed by flow cytometry. (**A**) Schematic diagram for functional analysis of BM-DCs maturation using rSIP. Cytometric Bead Array (CBA) assays on IL-6 (**B**), IFN-γ (**C**), TNF-α (**D**), and IL-10 (**E**) were shown in their respective panels. For these assays, day 6 BMDCs were treated with rSIP (1 and 10 μg/mL) and LPS (10 μg/mL) for 24 h. The culture supernatants of each group were subjected to CBA analysis by flow cytometry. Results represent 1 of 2 independent experiments. Results are shown as mean ± SEM. ** *p* < 0.01; **** *p* < 0.0001. BMDCs were treated with PBS-1X, rSIP (0,1; 1; 30 μg/mL), or LPS (1; 10 μg/mL) for 24 h; (**F**) cell viability and (**G**) cell death were determined using an Annexin V/PI staining. One representative result out of three similar experiments is shown. All bar graphs show the mean ± SD of four samples per group. **** *p* < 0.0001; ** *p* < 0.01; * *p* < 0.05 by ANOVA multiple comparisons for rSIP and LPS stimulation compared with PBS-1X. (**H**) BM-DCs were treated increment concentrations of rSIP purified by HPLC (0.1, 1, 10, and 30 μg/mL) and PBS-1X for 12 h. Numbers indicate percentage of BM-DCs positive for active caspase-3. NS = Not Significant. Results represent 1 of 2 independent experiments with similar results and are shown as mean ± SM.

**Figure 5 vaccines-08-00029-f005:**
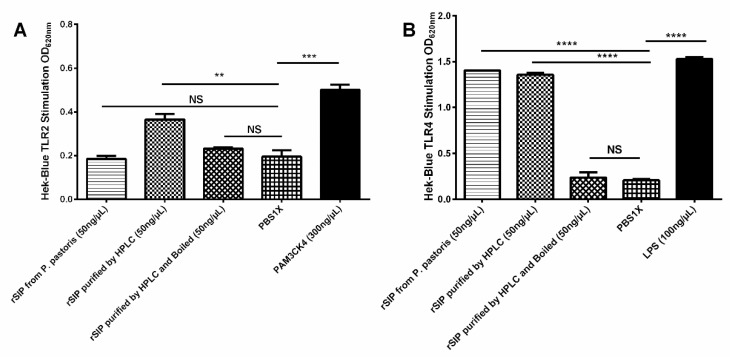
TLR2 and TLR4 reporter assays: HEK blue mTLR2 (**A**) and HEK blue mTLR4 (**B**) reporter cell lines were exposed for 24 h to purified rSIP from *P. pastoris*, purified rSIP from *E. coli*, rSIP degraded, and PBS-1X. The rSIP stimulates both TLR2 and TLR4. Values represent the mean of three independent experiments, and statistical significance was determined by a two-way ANOVA with a Bonferroni post-test (Not significant (ns); ** *p* < 0.01; *** *p* < 0.001; **** *p* < 0.0001).

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
