# Peer review of "Surface Immunogenic Protein of Streptococcus Group B is an Agonist of Toll-Like Receptors 2 and 4 and a Potential Immune Adjuvant"

_vaccines, 2020, doi:10.3390/vaccines8010029_

Round 1
Reviewer 1 Report
Summary: This manuscript details a study on the immune activating mechanisms of the Surface Immunogenic Protein from Group B Streptococcus for use as a potential vaccine adjuvant. It looked at the activation of bone marrow derived dendritic cells, cytokine expression, and TLR2/4 activation as well as increases in IgG antibodies in ovalbumin immunized mice.
Strengths:
Interesting use of SIP as a vaccine adjuvant. Ovalbumin is a well established immunodominant protein which is often found to induce immune responses even alone (as was shown by the authors) but it was interesting to see the increase in immune response by including SIP as an adjuvant. Also evaluated the mechanism of the response.
Weaknesses:
Major:
-The overall manuscript presentation needs to be revised, especially the flow of the introduction and conclusion needs to be improved. The grammar, overall, is not adequate, and the flow of the sections are difficult to follow. The conclusions about the use of this adjuvant in vaccines for pregnancy and children is confusing and unclear.
-The interpretation of the Th1 vs Th2 polarization is not completely correct. The authors state that both IL-12 and IL-10 are elevated which is consistent with a Th1 response. IL-10 has been shown to polarizes to a Th2 response (this is discussed in the citation doi:10.1007/s12250-015-3606-3) and also been shown to help suppress inflammation (reviewed: doi: 10.1084/jem.20070104). Further evaluation of the cytokine responses should be assessed such as IL-4, IL-2, and TFN-g responses or the interpretation of this data revised. Further, multiple studies have been published (and cited by this author) showing that the response to the SIP protein is both Th1 and Th2 (skewed to Th1 doi: 10.1016/j.molimm.2019.04.025).
-Figure 3: It is unclear from the results and figure description how the mean and standard deviation were determined. Were the three triplicates averaged and then the average and standard deviation reported or were all five mice averaged? Additionally, this figure could be improved by showing all five mice as individual values instead of the bar graph.
-Figure 4: This figure could be improved by describing exactly the morphologic change observed that the black arrows are pointing to in the description.
Minor:
Line 43: TR – missing L
Line 49: remove “the” from “the adjuvants”
Line 56: correct “fused to protein vaccines”
Line 60: “induces a humoral immune response” – while this may be a down stream immune response that occurs due to TLR2/4 activation, including this statement here in the list of other “innate immune responses” needs further explanation
Line 61: DC – first time this abbreviation is used
Lines 65-70 – these sentences are confusing; citation?
Line 72: “conservative” should be conserved
Line 74: Th1 responses are considered to be cell-mediated while Th2 are considered to be humoral responses. “Humoral Th1 immune response” should be revised.
Line 100: C57BL/6 is not enough information about the mouse strain used in this study due to the wide variation of divergence that has happened in this mouse strain over time. Where the strain was obtained from should be included.
Line 108: described where?
Line 173: sentence is confusing
Line 368: * is used to label the graph in Figure 3 but in not included in the description
Line 390: ** is used to label the graph in Figure 6 but in not included in the description
Author Response
Major comments and suggestions
The overall manuscript presentation needs to be revised, especially the flow of the introduction and conclusion needs to be improved. The grammar, overall, is not adequate, and the flow of the sections are difficult to follow. The conclusions about the use of this adjuvant in vaccines for pregnancy and children is confusing and unclear.Answer: As requested by the Reviewer, we modified the manuscript, correcting grammatical errors, and improving the English writing. Also, we improved the wording of the discussion of this article.
The interpretation of the Th1 vs Th2 polarization is not completely correct. The authors state that both IL-12 and IL-10 are elevated which is consistent with a Th1 response. IL-10 has been shown to polarize to a Th2 response (this is discussed in the citation doi:10.1007/s12250-015-3606-3) and also been shown to help suppress inflammation (reviewed: doi: 10.1084/jem.20070104). Further evaluation of the cytokine responses should be assessed such as IL-4, IL-2, and TFN-g responses or the interpretation of this data revised. Further, multiple studies have been published (and cited by this author) showing that the response to the SIP protein is both Th1 and Th2 (skewed to Th1 doi: 10.1016/j.molimm.2019.04.025).
Answer: As requested by the Reviewer, we modified and edited the manuscript to include the information relative to Th1 V Th2 polarization (lines 213-219). To improve the understanding of our results, we performed CBA assay to quantify in-vitro cytokine levels of BM-DCs pulsed with rSIP (Figure 4).
Figure 3: It is unclear from the results and figures description how the mean and standard deviation were determined. Were the three triplicates averaged and then the average and standard deviation reported or were all five mice averaged? Additionally, this figure could be improved by showing all five mice as individual values instead of the bar graph.
Answer: As requested by the Reviewer, we modified Figure and edited the legend (Figure 2, lines 494-506). In addition, in the new figure, we reported the values of five individual mice.
This figure could be improved by describing exactly the morphologic change observed that the black arrows are pointing to in the description.
Answer: As requested by the Reviewer and for a better understanding of our results, we decided to eliminate the figure. Also, in order to improve our manuscript, we decided to evaluate the pro-inflammatory state of dendritic cells pulsed with rSIP (Figure 4).
Minor comments and suggestions
Line 43: TR – missing L. Line 49: remove “the” from “the adjuvants”. Line 56: correct “fused to protein vaccines”. Line 60: “induces a humoral immune response” – while this may be a down stream immune response that occurs due to TLR2/4 activation, including this statement here in the list of other “innate immune responses” needs further explanation. Line 61: DC – first time this abbreviation is used. Lines 65-70 – these sentences are confusing; citation? Line 72: “conservative” should be conserved. Line 74: Th1 responses are considered to be cell-mediated while Th2 are considered to be humoral responses. “Humoral Th1 immune response” should be revised. Line 100: C57BL/6 is not enough information about the mouse strain used in this study due to the wide variation of divergence that has happened in this mouse strain over time. Where the strain was obtained from should be included. Line 108: described where? Line 173: sentence is confusing. Line 368: * is used to label the graph in Figure 3 but is not included in the description Line 390: ** is used to label the graph in Figure 6 but is not included in the description.Answer: As requested by the Reviewer, we modified the manuscript, correcting grammatical errors, and improving comments and suggestions.
We would like to thank the Reviewer again for their time and effort in reviewing this work. We hope that the current revised manuscript will be acceptable for publication in Vaccines.
Major comments and suggestions
The overall manuscript presentation needs to be revised, especially the flow of the introduction and conclusion needs to be improved. The grammar, overall, is not adequate, and the flow of the sections are difficult to follow. The conclusions about the use of this adjuvant in vaccines for pregnancy and children is confusing and unclear.Answer: As requested by the Reviewer, we modified the manuscript, correcting grammatical errors, and improving the English writing. Also, we improved the wording of the discussion of this article.
The interpretation of the Th1 vs Th2 polarization is not completely correct. The authors state that both IL-12 and IL-10 are elevated which is consistent with a Th1 response. IL-10 has been shown to polarize to a Th2 response (this is discussed in the citation doi:10.1007/s12250-015-3606-3) and also been shown to help suppress inflammation (reviewed: doi: 10.1084/jem.20070104). Further evaluation of the cytokine responses should be assessed such as IL-4, IL-2, and TFN-g responses or the interpretation of this data revised. Further, multiple studies have been published (and cited by this author) showing that the response to the SIP protein is both Th1 and Th2 (skewed to Th1 doi: 10.1016/j.molimm.2019.04.025).
Answer: As requested by the Reviewer, we modified and edited the manuscript to include the information relative to Th1 V Th2 polarization (lines 213-219). To improve the understanding of our results, we performed CBA assay to quantify in-vitro cytokine levels of BM-DCs pulsed with rSIP (Figure 4).
Figure 3: It is unclear from the results and figures description how the mean and standard deviation were determined. Were the three triplicates averaged and then the average and standard deviation reported or were all five mice averaged? Additionally, this figure could be improved by showing all five mice as individual values instead of the bar graph.
Answer: As requested by the Reviewer, we modified Figure and edited the legend (Figure 2, lines 494-506). In addition, in the new figure, we reported the values of five individual mice.
This figure could be improved by describing exactly the morphologic change observed that the black arrows are pointing to in the description.
Answer: As requested by the Reviewer and for a better understanding of our results, we decided to eliminate the figure. Also, in order to improve our manuscript, we decided to evaluate the pro-inflammatory state of dendritic cells pulsed with rSIP (Figure 4).
Minor comments and suggestions
Line 43: TR – missing L. Line 49: remove “the” from “the adjuvants”. Line 56: correct “fused to protein vaccines”. Line 60: “induces a humoral immune response” – while this may be a down stream immune response that occurs due to TLR2/4 activation, including this statement here in the list of other “innate immune responses” needs further explanation. Line 61: DC – first time this abbreviation is used. Lines 65-70 – these sentences are confusing; citation? Line 72: “conservative” should be conserved. Line 74: Th1 responses are considered to be cell-mediated while Th2 are considered to be humoral responses. “Humoral Th1 immune response” should be revised. Line 100: C57BL/6 is not enough information about the mouse strain used in this study due to the wide variation of divergence that has happened in this mouse strain over time. Where the strain was obtained from should be included. Line 108: described where? Line 173: sentence is confusing. Line 368: * is used to label the graph in Figure 3 but is not included in the description Line 390: ** is used to label the graph in Figure 6 but is not included in the description.Answer: As requested by the Reviewer, we modified the manuscript, correcting grammatical errors, and improving comments and suggestions.
We would like to thank the Reviewer again for their time and effort in reviewing this work. We hope that the current revised manuscript will be acceptable for publication in Vaccines.
Reviewer 2 Report
In this manuscript, authors observed that Surface Immunogenic Protein (SIP) is an immunomodulatory protein and a TLR2/TLR4 agonist. The manuscript also suggests that SIP could serve as a novel potential protein TLR agonist adjuvant and may be used in the development of new vaccines. The authors presented enough data to supportive of their conclusion. However, the study is largely limited by in vitro cell culture work, which makes hard to believe SIP could serve as an adjuvant in vivo.
Major points
It is unclear that authors only isolated bone marrow-derived DCs from female C57bl/6 mice. In Fig 2, it would be better to evaluate the inflammation status by detecting the cytokines/chemokines or immune cell counts in vivo. In Fig 3, the gender of C57bl/6 mice was not described. In Fig 4, SIP treatment significantly induced a morphological change in DCs. One concern is the cytotoxicity of SIP, authors should address in a dose-dependent manner.Author Response
Major comments and suggestions
It is unclear that authors only isolated bone marrow-derived DCs from female C57bl/6 mice.Answer: As suggested by the Reviewer, we added information regarding the experimental mice as part of the Materials and Methods section. In line with the Reviewer´s suggestion, our experiments were carried out in groups of five individuals (Figure 2).
In Fig 2, it would be better to evaluate the inflammation status by detecting the cytokines/chemokines or immune cell counts in vivo.
Answer: As requested by the Reviewer, we modified and edited the manuscript to include information relative to Figure 2 (lines 217-219). In Figure 2, we added serum cytokines levels (IL-4, IL-10, INF-γ, and IL-12) of mice immunized with PBS-1X, OVA, OVA + Alum, OVA+SIP. The cytokine determination suggests a systematic inflammatory state in mice immunized with rSIP, which could be related to the activation of TLR2 and TLR4.
In Fig 3, the gender of C57bl/6 mice was not described.Answer: As requested by the Reviewer, we have specified the gender of C57bl/6 mice in Materials and Methods (lines 77-86).
In Fig 4, SIP treatment significantly induced a morphological change in DCs. One concern is the cytotoxicity of SIP, authors should address in a dose-dependent manner.
Answer: As requested by the Reviewer, we eliminated Figure 4, and for a better understanding of our results, we evaluated proinflammatory cytokines and cytotoxicity of dendritic cells pulsed with rSIP (Figure 4).
We would like to thank the Reviewer again for their time and effort in reviewing this work. We hope that the current revised manuscript will be acceptable for publication in Vaccines.
Reviewer 3 Report
The manuscript entitled “Surface Immunogenic Protein of Streptococcus Group B is an Agonist of Toll-Like Receptor 2 and 4 and is a Potential Immune Adjuvant” by Diaz-Dinamarca et al., describes a study aimed at investigating the effect of the group B streptococcal Surface Immunogenic Protein (SIP) on antibody production in mice, dendritic cell maturation and activation as well as activation of TLR 2 and TLR 4 signaling in TLR reporter cells. Overall, the study question is interesting and the experimental design seems appropriate. Despite this, there are several concerns pertaining to the current manuscript and they are listed below.
Overall the manuscript must undergo extensive language editing. There are numerous sentences that are unclear and need rewording (e.g. lines 59-61, lines 68-70 etc.). The “Discussion” section contains numerous unclear statements and needs to be reworked. Moreover, abbreviations need to be defined (e.g. line 80: NI-NTA, line 86: BCA etc.). Line 77: Which GBS strain/species was used? Figure 3 legend indicates that it is S. agalactiae. Why use this particular streptococcal species? Are there any differences in SIPs derived from various streptococci? Why use these two different expression systems? As the results show, there are differences in function between SIPs expressed in E. coli versus P. pastoris. Please explain. Why did the investigators choose this particular immunization schedule? Please explain. Figure 2 indicates that mice were sacrificed six days after the last immunization (day 48). Line 100 states that they were euthanized five days later. In addition, please replace the word “challenge” as this would imply challenge with a pathogen. Line 99: Replace “post last immunization” with “of the experiment”. Line 110: Are these the same mice used in the vaccination experiment? Why on Day 0? Line 118: Please provide more detail. How and for how long were the cells “pulsed”? Line 130+: How were the cytokines detected (e.g. secondary conjugated antibody)? Please provide more information. Lines 145+: Were data assessed for normality? If yes, how? Please add information. Throughout “Materials and Methods” section: Please add information about controls (e.g. negative controls for flow cytometry and NF-kB activity assay). Line 154: Which protein (expressed in E. coli or P. pastoris)? Line 190: Is “antigenic” the correct term to use? Line 223: Incomplete sentence. Line 260: Foreign proteins can be potent immunogens with the potential for hypersensitivity reactions. Thus, the statement that protein vaccines are safe should be qualified. Figure 2 is superfluous. The information provided in this figure can be included in the “Materials and Methods” section. Figure 3: The authors describe the use of “Pre-immune” serum as a negative control. No information is provided in the Material and Methods section regarding the collection of mouse serum prior to immunization. Figure 4: These images are not very convincing. Please delete this figure. Information provided in the figure legend needs to be included in the “Material and Methods” section. Figure 5: Please include that PBS served as the negative control in the figure legend.Author Response
Major comments and suggestions
Overall the manuscript must undergo extensive language editing. There are numerous sentences that are unclear and need rewording (e.g. lines 59-61, lines 68-70 etc.). The “Discussion” section contains numerous unclear statements and needs to be reworked. Moreover, abbreviations need to be defined (e.g. line 80: NI-NTA, line 86: BCA etc.). Line 77: Which GBS strain/species was used?Answer: As requested by the Reviewer, we modified the manuscript, correcting grammatical errors, and improving the English writing. Also, we improved the wording of the discussion of the manuscript.
Reviewer. Figure 3 legend indicates that it is agalactiae. Why use this particular streptococcal species? Are there any differences in SIPs derived from various streptococci?
Answer: As requested by the Reviewer, we modified the manuscript, correcting Figure legend. In the context of the nature of the Surface immunogenic protein, this protein is only found in Group B Streptococcus. We use this protein because immunization using rSIP generates protection against GBS infection. In this sense, we wanted to evaluate its immune response as a protein vaccine adjuvant.
Reviewer. Why use these two different expression systems? As the results show, there are differences in function between SIPs expressed in coli versus P. pastoris. Please explain.
Answer: We used two different expression systems to prove that rSIP from E. coli activates TLR Cell-based Assay. Pichia pastoris was used as a rSIP expression system because it does not generate endotoxins that could generate a false positive in TLR activation assay (Figure 5). On the other hand, Pichia pastoris can generate glycosylation in the expressed protein, which could have an impact on the generation of less immunogenic capacity by the antigen, which would be associated with rSIP from Pastoris it is only a TLR4 agonist.
Reviewer. Why did the investigators choose this particular immunization schedule? Please explain.
Answer: We chose this particular immunization schedule because we wanted to follow the experiments of Liu et al., 2009; their work also describes an adjuvant protein with the capacity of potential immune adjuvant (lines 108-119).
Reviewer. Figure 2 indicates that mice were sacrificed six days after the last immunization (day 48). Line 100 states that they were euthanized five days later.
Answer: As suggested by the Reviewer, we added information regarding the experimental mice as part of the Materials and Methods section.
Reviewer. In addition, please replace the word “challenge” as this would imply challenge with a pathogen. Line 99: Replace “post last immunization” with “of the experiment”.
Answer: As requested by the Reviewer, we modified the manuscript and edited the English writing.
Reviewer. Line 110: Are these the same mice used in the vaccination experiment? Why on Day 0?
Answer: As requested by the Reviewer, we modified the description of the immunization schedule (Materials and Methods section, lines 108-119).
Reviewer. Line 118: Please provide more detail. How and for how long were the cells “pulsed”?
Answer: As requested by the Reviewer, we modified the description of the immunization schedule (Materials and Methods section, lines 136-138).
Reviewer. Line 130+: How were the cytokines detected (e.g. secondary conjugated antibody)? Please provide more information.
Answer: As recommended by the Reviewer, we improved the description of the methodology (lines 150-170).
Reviewer. Lines 145+: Were data assessed for normality? If yes, how? Please add information.
Answer: As recommended by the Reviewer, we improved the description of the methodology.
Reviewer. Throughout “Materials and Methods” section: Please add information about controls (e.g. negative controls for flow cytometry and NF-kB activity assay).
Answer: As recommended by the Reviewer, we modified the manuscript, correcting the Materials and Methods section (lines 180-184)
Reviewer. Line 154: Which protein (expressed in coli or P. pastoris)? Line 190: Is “antigenic” the correct term to use? Line 223: Incomplete sentence. Line 260: Foreign proteins can be potent immunogens with the potential for hypersensitivity reactions. Thus, the statement that protein vaccines are safe should be qualified.
Answer: As requested by the Reviewer, we modified and edited the manuscript to include information relative to these points.
Reviewer. Figure 2 is superfluous. The information provided in this figure can be included in the “Materials and Methods” section.
Answer: As recommended by the Reviewer, we modified the manuscript, correcting Figure 2.
Reviewer. Figure 3: The authors describe the use of “Pre-immune” serum as a negative control. No information is provided in the Materials and Methods section regarding the collection of mouse serum prior to immunization.
Answer: As requested by the Reviewer, we modified and edited the manuscript to include information relative to Pre-immune serum (lines 120-121).
Reviewer. Figure 4: These images are not very convincing. Please delete this figure. Information provided in the figure legend needs to be included in the “Materials and Methods” section.
Answer: As requested by the Reviewer, we modified and edited the manuscript. We deleted Figure 4, and in line with the recommendations of the other reviewers, we decided to evaluate proinflammatory cytokines and cytotoxicity in dendritic cells pulsed with rSIP (Figure 4).
Reviewer. Figure 5: Please include that PBS served as the negative control in the figure legend.
Answer: As requested by the Reviewer, we modified the description in the figure legend (lines 512-513)
We would like to thank the Reviewer again for their time and effort in reviewing this work. We hope that the current revised manuscript will be acceptable for publication in Vaccines.
Round 2
Reviewer 2 Report
Authors have addressed all my concerns satisfactorily.